Modic changes in patients with lumbar disc herniation followed more than 1 year after lumbar discectomy: a systematic review and meta-analysis

Feng Xiangyu 1
Nian Sunqi 2 niansunqi037@163.com
Chen Jiayu 2 3
http://orcid.org/0000-0003-2821-0547 Li Na 4
Duan Pingguo 5 ndyfy03692@ncu.edu.cn
1 Queen Mary School, Nanchang University , Nanchang, Jiangxi , China
2 Department of Orthopedics, The First People’s Hospital of Yunnan Province , Kunming, Yunnan , China
3 Department of Orthopedics, The Affiliated Hospital of Kunming University of Science and Technology , Kunming, Yunnan , China
4 Department of Anesthesiology, 920th Hospital of the Joint Logistics Support Force of the Chinese People’s Libration Army , Kunming, Yunnan , China
5 Department of Orthopedics, The First Affiliated Hospital of Nanchang University , Nanchang, Jiangxi , China
van Dieën Jaap
Electronic publication date: 2024 Aug 7
Publication date: 2024
Volume: 12
Electronic Location ID: e17851
Received 2024 Mar 12; Accepted 2024 Jul 11
Copyright: © 2024 Feng et al.
Copyright year: 2024
Copyright holder: Feng et al.
License: This is an open access article distributed under the terms of the Creative Commons Attribution License, which permits unrestricted use, distribution, reproduction and adaptation in any medium and for any purpose provided that it is properly attributed. For attribution, the original author(s), title, publication source (PeerJ) and either DOI or URL of the article must be cited.
License URL: https://creativecommons.org/licenses/by/4.0/

Keywords: Modic changes, Type conversion, Pain, Spine

Funding: The Yunnan Province Ten Thousand People Plan Young Talent Program YNWR-QNBJ-2019-184 The Reserve Talents for Academic and Technical Leaders of Middle-aged and Young People in Yunnan Province 202305AC160069 The Kunming University of Science and Technology School of Medicine Postgraduate Innovation Fund The Kunming University of Science and Technology Postgraduate Talent Program CA22369M099A This work was supported by The Yunnan Province Ten Thousand People Plan Young Talent Program (YNWR-QNBJ-2019-184), The Reserve Talents for Academic and Technical Leaders of Middle-aged and Young People in Yunnan Province (202305AC160069), The Kunming University of Science and Technology School of Medicine Postgraduate Innovation Fund, and The Kunming University of Science and Technology Postgraduate Talent Program (CA22369M099A). The funders had no role in study design, data collection and analysis, decision to publish, or preparation of the manuscript.

==============================
Background

Modic changes (MCs) are identified as an independent risk factor for low back pain. Different subtypes of MCs vary in their impact on postoperative pain relief. However, consensus on the transformation of postoperative MC fractions in patients with distinct MC subtypes is lacking.

Methods

This comprehensive systematic review and meta-analysis searched English-language articles in PubMed, Cochrane Library, Web of Science, and Embase databases until January 2024. Studies included focused on patients transitioning between various microcrack subtypes post-discectomy. The primary outcome measure was the transformation between different postoperative microcrack fractions.

Results

Eight studies with 689 participants were analyzed. Overall, there is moderate to high-quality evidence indicating differences in the incidence of MC conversion across MC subtypes. The overall incidence of MC conversion was 27.7%, with rates of 37.0%, 20.5%, and 19.1% for MC0, MC1, and MC2 subtypes, respectively. Thus, postoperative MC type transformation, particularly from preoperative MC0 to MC1 (17.7%) or MC2 (13.1%), was more common, with MC1 transformation being predominant. Patients with preoperative comorbid MC1 types (19.0%) exhibited more postoperative transitions than those with MC2 types (12.4%).

Conclusion

This study underscores the significance of analyzing post-discectomy MCs in patients with lumbar disc herniation, revealing a higher incidence of MCs post-lumbar discectomy, particularly from preoperative absence of MC to MC1 or MC2. Preoperative MC0 types were more likely to undergo postoperative MC transformation than combined MC1 or MC2 types. These findings are crucial for enhancing surgical outcomes and postoperative care.

Background

Modic changes (MCs) (Modic & Ross, 2007; Modic et al., 1988b) are commonly observed lesions in the vertebral endplates of the spine during radiological examinations. Scholars have proposed that Modic changes are independent risk factors for episodes of low back pain (Hopayian, Raslan & Soliman, 2023; Määttä et al., 2015). These changes manifest distinct pathological features and can be identified on magnetic resonance imaging (MRI). Modic type 0 (MC0) indicates no Modic changes. Modic type 1 (MC1) is characterized by bone edema, leading to reduced signal intensity on T1-weighted images and significant enhancement on T2-weighted images. In contrast, Modic type 2 (MC2) displays features of yellow fat with signal enhancement on both T1- and T2-weighted MRI images. Modic type 3 (MC3) represents sclerotic bone and demonstrates low signal intensity on both T1- and T2-weighted MRI images (Dudli et al., 2016; Modic et al., 1988a, 1988b).

Until the 2010s, follow-up studies of patients with MCs primarily focused on observing the natural progression of these changes, with limited emphasis on evaluating the impact of surgical interventions (Albert & Manniche, 2007; Jensen et al., 2009; Kuisma et al., 2006). Discectomy, a common medical procedure for severe disc herniation, has shown efficacy in patients who do not respond to long-term conservative treatment (Cenic & Kachur, 2009), especially in terms of reducing or eliminating low back pain in patients with various MC types (Nian et al., 2023). However, some articles have reported surgical promotion of Modic type transformation. For instance, Albert & Manniche (2007) noted a higher incidence of MCs among patients with low back pain who had undergone surgery within 1 year of follow-up than among those who did not undergo surgery. A systematic evaluation with higher-level evidence has shown that patients with preoperative MC1 tend to have poorer functional outcomes after discectomy, and MC after lumbar discectomy can significantly affect prognosis (Nian et al., 2023). While some studies have suggested a trend towards the conversion of MC0 and MC1 to MC2 (Albert & Manniche, 2007; Kuisma et al., 2006), a prospective cohort study pointed out a more complex pattern of MC transformation after limited discectomy in the lumbar spine (Bostelmann et al., 2019).

Therefore, understanding the impact of the type of MC alteration on pain and functional prognosis after discectomy is crucial. However, there is still a lack of high-level evidence regarding the postoperative transformation of MC alterations. This study aims to investigate the incidence and trend of MC transformation post-lumbar discectomy.

Methods

Study design

This review adhered to the MOOSE (Meta-analysis Of Observational Studies in Epidemiology) guidelines (Stroup et al., 2000) and was registered on PROSPERO (CRD42023465280), with an application submitted on September 29, 2023. Following registration, we initiated an initial search and pilot study selection process, followed by data extraction and analysis.

Search strategy

A comprehensive literature search was conducted across four databases: PubMed, Web of Science, Embase, and Cochrane up to January 2024. We used combinations of keywords such as “diskectomy or discectomy,” “Modic changes or VESC,” “Low back pain or LBP,” and “Lumbar disc herniation” in the fields “abstract” and “title.” The complete search strategy is detailed in Supplemental 1. Only published articles and human studies were included, with manual checks of references to maximize the number of documents cited.

Study selection

Inclusion criteria comprised studies where discectomy or microdiscectomy was performed for patients with lumbar disc herniation (LDH) combined with various Modic types. Additionally, studies needed to identify the type of MCs both preoperatively and postoperatively, including MC1 and MC2. Outcome indicators included preoperative and postoperative MC types. Both randomized controlled trials (RCTs) and non-RCTs studies were eligible, provided they were published in English. Exclusion criteria encompassed studies observing the natural evolution of MCs, those involving surgical procedures other than discectomy, those reporting incomplete follow-up data, and articles with non-extractable data such as case reports, reviews, or letters.

Data extraction

Two researchers independently reviewed and extracted data from included articles, including published authors, study year, type, duration, patient demographics, Modic type, surgical method, and outcome metrics at final follow-up. Disagreements were resolved through discussion. Data were formatted according to the Cochrane Handbook of Systematic Reviews (Higgins et al., 2019).

Risk of bias assessment

Two investigators evaluated all included studies. The quality of observational studies was assessed using the Newcastle-Ottawa Scale (NOS) (Bostelmann et al., 2019; Kawaguchi et al., 2023; Ohtori et al., 2010; Rahme et al., 2010; Takahashi et al., 2021; Yaman et al., 2017). The NOS scale (total score, 8), commonly employed as an evaluation tool for cohort studies, is composed of three parts: Selection of the study population, comparability between groups, and outcome measures. In the study population selection, one point was rated for meeting the representativeness of the exposed group, the method of selecting the nonexposed group, the method of determining exposure factors, and the determination of outcome indicators at the beginning of the study; two points for controlling for confounding factors during design and statistical analysis; and one point each for adequacy of the assessment of study outcomes, follow-up time, and follow-up in the outcome measures section. Table 1 presents detailed components of the NOS scale. Figure 1 illustrates the Risk of Bias (RoB2) tool used for the risk of bias evaluation for RCTs (el Barzouhi et al., 2014; Lee & Bae, 2015). Risk of bias may arise from various factors, including the country of study, selection of the study population, choice of procedures, and duration of follow-up. Disagreements, if any, were resolved through consultation with an independent third party.

Table 1 The detailed description of newcastle-ottawa scale (NOS) used to evaluate included non-RCT articles.

	Selection	Comparability	Outcomes		
Study	A	B	C	D	E	F	G	H	Score	
Takahashi et al. (2021)	▲	▲	▲	▲	▲	▲	▲	▲	8	
Bostelmann et al. (2019)	▲	▲	▲	▲	▲▲	▲	▲	▲	9	
Yaman et al. (2017)	▲	▲	▲		▲	▲	▲	▲	7	
Rahme et al. (2010)	▲	▲	▲	▲	▲▲	▲	▲	▲	9	
Ohtori et al. (2010)	▲	▲	▲			▲	▲	▲	6	
Kawaguchi et al. (2023)	▲	▲	▲	▲	▲▲	▲	▲	▲	9	
Note:

A, Representativeness of the exposed cohort; B, Selection of the non-exposed cohort; C, Ascertainment of exposure; D, Demonstration that outcome of interest was not present at start of study; E, Comparability of cohorts on the basis of the design or analysis; F, Assessment of outcome; G, Follow-up long enough for outcomes to occur; H, Adequacy of follow up of cohort.

Figure 1 The RoB2 tool utilized for risk of bias assessment of two RCT articles.

Each point illustrates the average performance of all included articles on the risk of bias corresponding to each section. Green indicates the low risk meanwhile the yellow indicates the unclear risk.

Data analysis

Post-surgery, MC types were categorized into MC0, MC1, and MC2 subtypes, and their conversion rates between groups were measured. Specific conversions (MC0→MC1, MC2 and MC3; MC1→MC0, MC2 and MC3; MC2→MC0, MC1 and MC3) between MC types were analyzed, including extreme values, which implies that both scenarios involving a complete set of conversions during measurements and those without any conversions are encompassed. The above steps are performed using the “metaprop” command in the Stata 17® software with associated meta-analysis plug-ins. Preoperative MC3 and mixed MC types were excluded due to sample size limitations and incomplete data. This study aimed to analyze MC conversion rates between different MC types as outcome metrics. We used preoperative and postoperative MCs as indicators, setting a 95% confidence interval (95% CI). Additionally, we conducted a meta-analysis of preoperative and postoperative MCs to compare the proportion of MC conversions across MC0, MC1, and MC2 groups. Meta-analysis was performed using a random-effects model based on data heterogeneity (I2 > 50%).

Statistical analysis

The study followed the MOOSE statement guidelines, with literature processing managed using Endnote X9 software (https://proquest.libguides.com/endnote/home). Meta-analysis was conducted using Review Manager Software (https://training.cochrane.org/online-learning/core-software/revman).

Results

Study selection

Initially, we identified 100 articles, which included 13 articles from PubMed, 18 articles from Embase, 63 articles from Web of Science, four articles from Cochrane, and two articles from manual search. After removing duplicates, 58 articles remained. Through title and abstract screening, 39 articles were further excluded, leaving 19 articles for full-text assessment. Subsequently, six studies were excluded due to no Modic type illustration, four for incomplete outcomes, and one for no clear surgical procedure, resulting in eight articles (Bostelmann et al., 2019; el Barzouhi et al., 2014; Kawaguchi et al., 2023; Lee & Bae, 2015; Mostofi, Moghaddam & Peyravi, 2018; Ohtori et al., 2010; Rahme et al., 2010; Takahashi et al., 2021; Yaman et al., 2017) included in the meta-analysis (Fig. 2).

Figure 2 Study flow diagram follow PRISMA guidelines.

Study characteristics

Table 2 provides detailed characteristics of the included studies, including country of origin, patient demographics, procedure types, body mass index, duration of pain, MC groups, follow-up time, and MC diagnosis criteria. All studies utilized similar MC diagnosis criteria and grouping standards. Although some articles mentioned inclusion criteria for MC3 and mixed MC types (Lee & Bae, 2015), these subtypes were excluded due to limited data. All eight studies presented outcome indicators at both preoperative and postoperative time points. In total, 728 patients diagnosed with LDH were identified, with 689 patients meeting the inclusion criteria (445 with MC0 (64.6%), 70 with MC1 (10.2%), and 174 with MC2 (25.2%)), after excluding cases with preoperative MC3, mixed MC types, or lost follow-up. The incidence of MC conversion in all groups was 27.7%. Patients with MC0 exhibited a higher incidence of conversion to any other MC types (37.0%), compared to those with MC1 and MC2 (20.5% and 19.1%, respectively; Figs. 3–6).

Table 2 Summary of eight included articles.

Author (Publish year)	Country	Journal	Study period	Procedure type	Age	BMI	Position of Modic change	Female (%)	Modic type before surgery	Modic type after surgery	Clinical symptoms	Duration of pain	Follow-up period (baseline)	Inclusion criteria	Exclusion criteria	MC diagnosis	
Prospective study	
Takahashi et al. (2021)	Japan	BMC Musculoskeletal Disorders	2010–2018	Microdiscectomy	43.6 + 14.7 (<75)	/	L3/4:6	31 (48%)	MC0:48	MC0:40	Persistent and unremitting LBP	>3 months	1 year	(1) The presence of persistent and unremitting LBP for more than 3 months, X-ray images, and magnetic resonance imaging (MRI).	(1) Patients with LSS or degenerative spondylolisthesis (DS) comorbidities.	Type 1 showed decreased signal intensity on T1-weighted images and increased signal intensity on T2-wcighted images.	
MC1:3	MC1:8	(2) Patients who were diagnosed with a lateral herniation treated with fusion surgery	
L4/5:28	MC2:10	MC2:12	(2) Those who were not improved by sufficient conservative treatment and wished to undergo surgical treatment.	(3) Patients who were with thoracic myelopathy and hip osteoarthritis.	Type 2 showed increased signal intensity on T1-weighted images and isointense or slightly increased signal intensity on T2-weightcd images (Disc degeneration was graded using the Pfirrmann scale.)	
L5/S:31	MC3:1	MC3:2	(4) Patients with recurrent herniation.	
Unable:3			
Rahme et al. (2010)	Lebanon	Journal of Neurosurgery-Spine	2004–2008	Microdiscectomy	Mean:54 (24–78)	/	L4-L5 and L5-S1	14 (34%)	MC0:22	MC0:9	/	/	3–5 years	Patients with a virgin (previously nonoperated) lumbar spine.	/	1) Type 1 changes, which are hypointense on T1-weighted imaging and hyperintense on T2-weighted imaging and represent bone marrow edema and inflammation;	
									MC1:5	MC1:6						2) Type 2 changes, which are hyperintense on T1-weighted imaging and isointense or slightly hyperintense on T2-weighted imaging and are associated with fatty degeneration of the bone marrow;	
									MC2:14	MC2:26							
Ohtori et al. (2010)	Japan	Spine (Phila Pa 1976)	/	Discectomy	35.5 ± 5	/	L4-L5:27	19 (42.2%)	MC0:22	MC0:19	LBP and leg pain	>3 months	2 years	(1) Patients had low back and leg pain, continuing for at least 3 months.	(1) Patients who had previously undergone spinal surgery	Modic type 1 signals (low intensity on T1weighted spin-echo images and high intensity on T2-weighted spin-echo images), those with Modic type 2 signals (high intensity on both T1- and T2-weighted spin-echo images), and those with	
MC1:21	
L5-S1:18	MC1:23	MC2:6	(2) Patients were diagnosed with lumbar disc herniation on MRI.	(2) Spinal tumor, infection, and trauma	
MC3:0	
Retrospective study																
Kawaguchi et al. (2023)	Japan	Global Spine Journal	Between January 2013 and December 2018	Microscopic discectomy	47.4 ± 10.2	23.2 ± 2.7	L3/4: 6	40 (41%)	MC0:96	MC0:67	Redicular pain	11.6 ± 5.3 wks	18 ± 4.6 months	The inclusion criteria were lumbar disc herniation on MRI with corresponding radicular pain, failure of conservative treatment for 6 weeks or more, no spinal instability as assessed on functional radiographs, and a minimum 1-year follow-up following discectomy.	The exclusion criteria were coexisting lumbar spinal canal stenosis, history of lumbar surgery, cauda equina syndrome, and insufficient collection of tissue samples	Modic change at the operated level was assessed based on Modic classification which was divided into 6 groups; type 0, type 1, type 2, type 3, and mixed type 1/2 and 2/3	
MC1:6	
L4/5:49	MC2:14	
L5/S:41	MC2:14	
Bostelmann et al. (2019)	Germany	European Spine Journal	2010–2014	Discectomy	44.0 + 7.0 (21–75)	25.78 (23.23–29.22)	/	107(38%)	MC0:115	MC0:69	Back pain/Leg pain/Combined	>6 wks	2 year	(1) Age between 21 and 75 years	(1) Prior index level surgery.	Type 0: No oedematous reaction or vascular congestion induced in the adjacent bone marrow of the endplates	
MC1:30	MC1:64	(2) Spondylolisthesis with >25% slip.	Type 1: Hypointense reaction and vascular congestion in the adjacent marrows on T1-weighted MR imaging; hyperintense on T2-weighted images, new or increased relative to the previous time point	
MC2:121	MC2:96	(2) The presence of lumbar radiculopathy due to a primary, single-level disc herniation between L1 and S1	(3) Scoliosis of greater than 10 degrees (both angular and rotational).	Type 2: Bone marrow converted to a predominantly fatty marrow. Hyperintense on T1 and isointense to hypointense on T2. The exact signal intensity depends on the degree of T2 weighting	
MC3:1	MC3:2	(3) Failure of at least 6 weeks of nonsurgical treatment.	(4) Insulin-dependent diabetes.		
	Unable:36	(5) Contraindication for MRI.		
Yaman et al. (2017)	Turkey	Hong Kong Med J	2004–2009	Discectomy	53.2 ± 12.3	24.8 ± 0.8	L4-L5	8 (32.0%)	MC0:7	MC0:5	Radicular pain with or without neurological deficit, numbness in the lumbar spine, buttock, and/or lower extremity	>3 months	Average 320 days	(1) Radicular pain for at least 3 months that was refractory to 6 weeks of conservative treatment with or without neurological deficit,	(1) Prior lumbar surgery at another institution	Modic endplate changes were also classified with the help of their radiological department on T1/T2weighted sagittal MRI sequences.	
MC1: 4	MC1:6	(2) Numbness in the lumbar spine, buttock, and/ or lower extremity,	(2) Segmental instability,	
(3) Vertebral fractures and spinal infections,	
MC2:11	MC2:11	(3) Age between 21 and 75 years,	(4) Other types of degenerative disc disease	
(5) Tumours	
MC3:3	MC3:3	(4) Magnetic resonance imaging (MRI) and/or computed tomography demonstrating anatomical unilateral LDH correlating with symptoms.	(6) Pregnancy,	
(7) Age over 75 years	
Random controlled trial																
el Barzouhi et al. (2014)	Netherlands	The Spine Journal : Official Journal of the North American Spine Society	/	Discectomy	41.9 + 9.8	26.2 ± 3.9	L3-L4:7	57 (34%)	MC0:99	MC0:32	Persistent radicular pain in the L4-S1 dermatome with or without neurological deficit/Severe disabling leg pain	6-12 wks	1 year	(1) Age 18–65 yr	(1) Cauda equina syndrome or severe paresis (MRC < 3)	Type 1 lesions, hypointense on T1-weighted images and hyperintense on T2-weighted images, represent marrow edema, and are associated with an acute process. Type 2 lesions, the most common type, have increased signal on T1-weighted images and isointense or slightly hyperintense signal on T2-weighted images, and represent fatty degeneration of subchondral marrow and are associated with a chronic process.	
MC1:82	(2) Persistent radicular pain in the L4, L5 or S1 dermatome with or without mild neurological deficit	(2) Complaints of a lumbosacral radicular syndrome in the same dermatome within the past 12 months	
MC1:1	MC2:31	(3) Severe disabling leg pain of 6–12 weeks duration	(3) A history of unilateral disc surgery on the same level	
L4-L5:73	Mixed MC1 and MC2:20	(4) Evidence of a unilateral disc herniation confirmed on MRI	(4) Spinal canal stenosis	
MC2:67	Otherwise:3	(5) Sufficient knowledge of Dutch language	(5) Degenerative or lytic spondylolisthesis	
	(6) Informed consent	(6) Pregnancy	
L5-S1:88	Mixed MC1 and MC2:1			(7) “Severe life-threatening” or psychiatric illness	
		(8) Planned migration to another country in the year after randomization	
Lee & Bae (2015)	South Korea	Int J Clin Exp Med	2008–2009	Automated open lumbar discectomy (AOLD) or Microdiscectomy (MD)	42.7 ± 11.5	/	L3-4/L4-5/L5-S1 1/14/25	13 (32.5%)	MC0:24	MC0:18	Pain/Neurologic deficit	4.6 ± 4.9 Wks	20 months	(1) The presence of disc herniation as determined by magnetic resonance imagining (MRI) and pain that persisted for 4-8 weeks after conservative treatment involving rest, analgesia and physical therapy.	(1) Patients age older than 69 years, previous surgery, severe lumbar stenosis, spondylosis, spondylolisthesis, extraforaminal far lateral disc herniation, foraminal spur or bony compression, metabolic bone disease,	Type 1 Modic changes were hypointense on T1-weighted imaging (T1WI) and hyperintense on T2-weighted imaging (T2WI), type 2 Modic changes were hyperintense on T1WI and isointense or slightly hyperintense on T2WI, and type 3 Modic changes were hypointense on both T1WI and T2WI.	
MC1:1	MC1:7	
MC2:3	MC2:3	(2) Patients with progressive neurologic deficit underwent emergency operations.	(2) Patients receiving worker’s compensation, and patients who have coexisting lumbar spinal disease.	
MC3:0	MC3:0	

Figure 3 Meta-analysis results for MC conversion more than 1 year after discectomy in all groups.

Red dotted line points out the overall proportion of Meta-analysis. MC, Modic changes; CI, Confidence interval; ES, Effect size (Kawaguchi et al., 2023; Takahashi et al., 2021; Yaman et al., 2017; el Barzouhi et al., 2014; Bostelmann et al., 2019; Lee & Bae, 2015; Ohtori et al., 2010; Rahme et al., 2010).

Figure 4 Meta-analysis results for MC0 conversion towards MC1, MC2 and MC3 groups more than 1 year after discectomy.

MC, Modic changes; CI, Confidence interval; ES, Effect size (Kawaguchi et al., 2023; Takahashi et al., 2021; Yaman et al., 2017; el Barzouhi et al., 2014; Bostelmann et al., 2019; Lee & Bae, 2015; Ohtori et al., 2010; Rahme et al., 2010).

Figure 5 Meta-analysis results for MC1 conversion towards MC0, MC2 and MC3 groups more than 1 year after discectomy.

MC, Modic changes; CI, Confidence interval; ES, Effect size (Kawaguchi et al., 2023; Takahashi et al., 2021; Yaman et al., 2017; el Barzouhi et al., 2014; Bostelmann et al., 2019; Lee & Bae, 2015; Ohtori et al., 2010; Rahme et al., 2010).

Figure 6 Meta-analysis results for MC2 conversion towards MC0, MC1 and MC3 groups more than 1 year after discectomy.

MC, Modic changes; CI, Confidence interval; ES, Effect size (Kawaguchi et al., 2023; Takahashi et al., 2021; Yaman et al., 2017; el Barzouhi et al., 2014; Bostelmann et al., 2019; Lee & Bae, 2015; Ohtori et al., 2010; Rahme et al., 2010).

Risk of bias assessments

All eight studies underwent NOS or RoB2 evaluation, with none deemed to be at high risk of bias. The scores varied, with three studies rated eight points, two studies rated eight points, one study rated seven points, and one study rated six points. After completing the methodology quality assessment, the results showed acceptable quality.

Preoperative MCs

Data extraction from all eight studies allowed for the analysis of MC conversion incidence between MC0, MC1, and MC2 groups. Figure 3 depicts forest plots for the incidence of MC conversion across these groups. The conversion rate from MC0 (37.0%, p < 0.0001, 95% CI: [20.4–55.1]) was higher than that from MC1 (20.5%, p = 0.258, 95% CI [7.2–36.8]) and MC2 (19.1%, p < 0.0001, 95% CI [0.6–49.0]), as well as the overall incidence (27.7%, p < 0.0001, 95% CI [16.3–40.4]). However, high heterogeneity was observed (MC0, I2 = 92.063%; MC1, I2 = 22.405%; MC2, I2 = 90.205%; overall, I2 = 87.361%).

Postoperative MCs

Meta-analysis of the conversion rates towards MC0, MC1, and MC2 groups from all included studies was conducted (Fig. 4–7). Conversion rates towards MC0 from both MC1 and MC2 groups, as well as those towards MC3 across all groups, approached 0% (MC1→MC0, 0%, p = 0.969, 95% CI [0–1.3]; MC2→MC0, 0.8%, p = 0.113, 95% CI [0–6.7]; MC0→MC3, 0.1%, p = 0.849, 95% CI [0–0.1]; MC1→MC3, 0%, p = 0.959, 95% CI [0–0]; MC2→MC3, 0%, p = 0.613, 95% CI [0–1.6]). MC0 group had a higher proportion of conversion to MC1 than the MC2 group (MC0→MC1, 17.7%, p < 0.001, 95% CI [6.8–31.7]; MC2→MC1, 12.4%, p < 0.001, 95% CI [0.4–32.7]). Conversely, the MC1 group showed a higher tendency to convert into MC2 than the MC0 group (MC0→MC2, 13.1%, p < 0.001, 95% CI [6.7–21.0]; MC1→MC2, 19.0%, p = 0.329, 95% CI [7–33.8]). Moderate to high heterogeneity persisted across most subgroups (MC0→MC1 and MC2; I2 = 89.689% and 73.367%, respectively; MC1→MC2; I2 = 13.159%; MC2→MC0 and MC1: I2 = 43.911% and 82.178%, respectively).

Figure 7 Summary of percentage of Modic changes transformation between different MC types.

Discussion

To the best of our knowledge, this review presents the first examination of long-term outcomes (more than 1-year post-surgery) regarding MC transformations in patients with LDH, particularly those with a mix of preoperative MC types. The evidence, of moderate to high quality, indicates a higher likelihood of Modic type transitions in patients initially without preoperative MC (MC0) than in those with pre-existing combined MC types (MC+). Specifically, a significant proportion of patients transitioned to MC1 (17.7%), followed by MC2 (19.0%), while transitions to MC3 were notably infrequent (0%). Interestingly, individuals with a preoperative blend of MC1 types were more prone to such transformations than those with MC2 phenotypes, although the prevalence were similar (MC1 = 20.5%, MC2 = 19.1%). However, neither group showed evidence of reverting back to MC0. The process illustrating the conversion of Modic changes is depicted in Fig. 8.

Figure 8 Diagram of conversion between MC subtypes.

On the T1 sequence, MC1 presents a low signal and MC2 presents a high signal. Sources: https://doi.org/10.1016/j.spinee.2022.10.008.

Previous research indicates that patients with LDH exhibiting combined preoperative MC1 are associated with diminished functional status post-discectomy. Traditionally, prognostic assessments post-discectomy have focused on preoperative MCs (Karadağ et al., 2022; Li et al., 2021; Ulrich et al., 2020; Xu et al., 2019), yet the impact of surgical intervention on MC trajectory has not been thoroughly examined. The transformation of MCs following surgery may significantly contribute to persistent postoperative pain, underscoring the importance of analyzing the prevalence and patterns of MCs post-lumbar discectomy for understanding surgical outcomes and refining clinical management strategies. Furthermore, the progression of Modic changes post-discectomy is clinically significant as it can influence long-term outcomes and pain management strategies in patients with lumbar disc herniation (Albert & Manniche, 2007; Jensen et al., 2012; Kuisma et al., 2007). A comprehensive understanding of these changes can aid in tailoring postoperative care and potentially enhancing patient prognosis.

In our aggregate analysis, the preoperative incidences of MC0, MC1, and MC2 were found to be 37.0%, 20.5%, and 19.1%, respectively. Importantly, our study’s MC prevalence substantially exceeded the rates observed in the general Danish population’s natural progression (Jensen et al., 2009). This disparity may be attributed to the strong association between disc degeneration and MCs. Evidence from various studies supports a positive association between the emergence of MCs and progression of lumbar disc degeneration (Kjaer et al., 2005), with severe lumbar disc degeneration being more prevalent in patients exhibiting MCs (Özcan-Ekşi et al., 2021).

Recent scholarly efforts advocate for a nuanced classification of MCs, such as Xu et al. (2016)’s observation of mixed types combining endplate edema and sclerosis in MC patients. However, our study focused on traditional MC typology due to limited representation of mixed-type cases, potentially increasing heterogeneity and confounding factors in our findings. It is widely accepted that MCs can evolve from one type to another, representing different phases of the same pathological process. The concept of mixed types likely represents a transitional phase (Weishaupt et al., 1998), with evidence suggesting surgical interventions may accelerate this transition (Albert et al., 2008; Kanna et al., 2017), contrasting with the average follow-up duration in our included studies.

Moreover, Bostelmann et al.’s (2019) study, which met our study’s inclusion criteria, examined a population with low back pain persisting for just over 6 weeks, notably shorter than that in all included studies. This brief duration may indicate a higher prevalence of unstable MC1 in patients, potentially leading to more frequent MC transitions post-surgery. In eight studies, patients who underwent lumbar discectomy exhibited conversions between MC1 and MC2, with a higher incidence of transitioning from MC1 to MC2 preoperatively than from MC2 to MC1. Notably, no conversion to the more stable MC3 type was observed post-surgery, possibly due to the brief follow-up timeframe, highlighting the need for extended follow-up in future research.

Strengths and limitations

We conducted a comprehensive literature review and screening process spanning from 1998 to 2024, adhering to stringent MOOSE guidelines (Stroup et al., 2000). Our synthesized findings, supported by postoperative follow-up data, provide a robust foundation for evaluating the implications of MCs in postoperative scenarios. However, the study is not without limitations. First, the number of relevant articles included was limited, potentially resulting in a significant deviation from the actual MC conversion rate. Second, this study specifically focuses on patients who are more than 1 year post-surgery; therefore, changes that occur within 1 year after surgery remain unknown. Third, the dynamic nature of MC conversion, combined with varying follow-up durations post-discectomy, complicated our understanding of the frequency and timing of such transformations. Fourth, variations in MC types at different follow-up intervals remain unclear. Finally, the scarcity of data in the selected literature limits our insight into the precise locations of MCs and their postoperative implications on adjacent spinal segments.

Conclusion

Our findings suggest an increased likelihood of postoperative MC transformation, particularly from no preoperative MC (MC0) to MC1 or MC2, with a higher prevalence of MC1. Notably, patients exhibiting a combination of preoperative MC1 types experienced more frequent postoperative transformations than those with MC2 types. This study elucidates the characteristic changes in Modic change (MC) types observed among patients undergoing lumbar discectomy, thereby enhancing comprehension of surgical outcomes and facilitating improvements in postoperative clinical care quality.

Supplemental Information

Supplemental Information 1 Search strategy.

Supplemental Information 2 PRISMA checklist.

Supplemental Information 3 Raw data.

The case number and modic types of all included articles.

List of abbreviations

MCs Modic changes

Modic type 0/1/2/3 MC0/MC1/MC2/MC3

MRI Magnetic Resonance Imaging

LDH Lumbar disc herniation

RCTs Random controlled trials

95% CI 95% confidence interval

Additional Information and Declarations

Competing Interests

Author Contributions

Data Availability

The authors declare that they have no competing interests.

Xiangyu Feng performed the experiments, analyzed the data, prepared figures and/or tables, authored or reviewed drafts of the article, and approved the final draft.

Sunqi Nian conceived and designed the experiments, performed the experiments, analyzed the data, authored or reviewed drafts of the article, and approved the final draft.

Jiayu Chen conceived and designed the experiments, authored or reviewed drafts of the article, and approved the final draft.

Na Li conceived and designed the experiments, authored or reviewed drafts of the article, and approved the final draft.

Pingguo Duan performed the experiments, authored or reviewed drafts of the article, and approved the final draft.

The following information was supplied regarding data availability:

This is a systematic review/meta-analysis.

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
