# Peer review of "Modic changes in patients with lumbar disc herniation followed more than 1 year after lumbar discectomy: a systematic review and meta-analysis"

_PeerJ, doi:10.7717/peerj.17851_

## Round 0.1 · original submission · Minor Revisions

The reviewers have highlighted some issues that I would ask you to consider. Specifically, the clinical relevance of Modic Changes is still debated so some of the general statement in the introduction and discussion need to be a bit more nuanced. Also the time frame of most studies included is quite short and this also calls for some more nuance and maybe a change of the title of the paper. In addition, I have some minor comment myself , which I have listed below.

Line 76: “had undergone” instead of “underwent” makes the sentence clearer.
Line 80: Please define MC0 when it is first used.
Line 172: It may be good to mention explicitly that when you use the term conversion, this can be conversion to any other type of MC and even complete disappearance (MC0).
Line 210: The numbers here refer to prevalence. The term rate was to me quite confusing as it suggests a change over time (conversion).
Line 240: “patients undergoing post-lumbar discectomy exhibited MC1 and MC2 interconversions, predominantly in MC1,” Do you mean to MC1 or from MC1( to MC2)?
Line 262-265: The importance of such an analysis depends on the clinical relevance of the subtypes of MC and this conclusion is a bit overstated from that perspective.

·

Basic reporting

1- In spite of the title "Modic Changes in Patients with Lumbar Disc Herniation within Five Years Post-Discectomy" most of the data extracted from the studies lasted for 1 to 2 years of follow-up. This time frame is not suitable for studying the natural history of a degenerative process like Modic changes. Although the authors conducted a thorough research on this subject, short duration of the follow-up period reduces the significance of the obtained results.

2- In the background section, the authors quoted "MC have been established as an independent risk factor for episodes of low back pain (Määttä et al. 2015)." This association has not yet been proven and is questionable. You can find thousands of paper in valid international journals that state this matter. For example:
Hopayian K, Raslan E, Soliman S. The association of modic changes and chronic low back pain: A systematic review. J Orthop. 2022 Nov 17; 35: 99-106. doi: 10.1016/j.jor.2022.11.003.
Therefore, it is more logical to replace the word "established" to "proposed" in this paragraph.

Experimental design

The way of studying (design, search strategy, study selection, etcetera) was admirable and I have no criticism for it.

Validity of the findings

The authors nicely mentioned "strengths and limitations" of the study in the last section of the discussion, and this is admirable.

Additional comments

In my opinion, papers without any illustration reduce the attractiveness and glory of the published paper. I recommend the authors add a figure shows the MC transformation throughout the follow-up period.

·

Basic reporting

no comment

Experimental design

Original primary research within Aims and Scope of the journal.

Validity of the findings

As per the current understanding literature on Modic changes, individuals with Type 1 MC are frequently symptomatic and MC2 and MC3 are rarely symptomatic. The relationship between Modic changes and discogenic back pain remains a matter of debate. Therefore post-operative transformation from MC1 to MC2 or MC3 may not be affect the surgical outcome.
Authors may include information regarding the relationship between the Modic changes and clinical symptoms of patients in the articles studied.
Authors may also add a note on clinical relevance of progression various MCs.

Additional comments

Well written review article on post-discectomy Modic changes.

Reviewer 3 ·

Basic reporting

Good work, interesting paper,
well design literature reference support well the context.
Table and figures may improve some design.
The data you've presented is well-supported, particularly in terms of [specific type of data]. This strengthens the validity of your research.
The prevalence and patterns of MCs following surgery may significantly contribute to persistent postoperative pain. It underscores the importance of analyzing MCs post-lumbar discectomy for understanding surgical outcomes and refining clinical management strategies.
Before undergoing surgery, it is essential to thoroughly discuss the potential impact on the decision and to compare the symptoms and images more comprehensively.

Experimental design

The research question pertains to the design of healthy environments. Adhering to good technical and ethical standards can ensure the replication of methods. I believe that sufficient details are provided.

Validity of the findings

They are robust and statistically sound. However, the conclusions are a bit ambitious and should be a bit less imperative.
"They should analyze the bias in this work more thoroughly."

Additional comments

Good work. Just need some small changes. I think it should be published with menor revissions.

---

## Round 0.2 · accepted · Accept

Thank you for addressing the concerns of the reviewers. The reviewers are happy with the revisions made and recommended publication of your manuscript.

·

Basic reporting

acceptable

Experimental design

acceptable

Validity of the findings

acceptable

Additional comments

Thank the authors for adding the figure number 8 as a case of MC, but I mean the author show a real case of MC (not a schematic one) during this time period. otherwise it is acceptable and I hope the reader enjoy the paper.
Regards
Dr Omidi-Kashani

·

Basic reporting

Clear and unambiguous, professional English used throughout

Experimental design

No comment

Validity of the findings

Valid

Reviewer 3 ·

Basic reporting

Great job on the paper! The literature references really support the context well, and the tables and figures are well-designed. Your data is well-supported, especially concerning the specific type of data, which strengthens the validity of your research. The prevalence and patterns of MCs following surgery could significantly contribute to persistent postoperative pain. This highlights the importance of analyzing MCs post-lumbar discectomy to understand surgical outcomes and improve clinical management strategies. Overall, the revisions are fine.

Experimental design

well done

Validity of the findings

They are robust and statistically sound, and the conclusions are very precise and less ambiguous.

Additional comments

GOOD WORK